

# FERDCNN: an efficient method for facial expression recognition through deep convolutional neural networks

Metwally Rashad[1,2], Doaa Alebiary[1], Mohammed Aldawsari[2], Ahmed Elsawy[1,3] and Ahmed H. AbuEl-Atta[1]

[1] Faculty of Computers and Artificial Intelligence, Benha University, Benha, Egypt
[2] Department of Computer Engineering and Information, College of Engineering, Prince Sattam Bin Abdulaziz University, Al-Kharj, Saudi Arabia
[3] Information Technology Department, Faculty of Technological Industry and Energy, Delta Technological Unversity, Egypt, Egypt

## ABSTRACT

Facial expression recognition (FER) has caught the research community's attention recently because it can affect many real-life applications. Multiple studies have focused on automatic FER, most of which use a machine learning methodology, FER has continued to be a difficult and exciting issue in computer vision. Deep learning has recently drawn increased attention as a solution to several practical issues, including facial expression recognition. This article introduces an efficient method for FER (FERDCNN) verified on five different pre-trained deep CNN (DCNN) models (AlexNet, GoogleNet, ResNet-18, ResNet-50, and ResNet-101). In the proposed method, firstly the input image has been pre-processed using face detection, resizing, gamma correction, and histogram equalization techniques. Secondly, the images go through DCNN to extract deep features. Finally, support vector machine (SVM) and transfer learning are used to classify generated features. Recent methods have been employed to evaluate and contrast the performance of the proposed approach on two publicly standard databases namely, CK+ and JAFFE on the seven classes of fundamental emotions, including anger, disgust, fear, happiness, sadness, and surprise beside neutrality for CK+ and contempt for JAFFE. The suggested method tested Four different traditional supervised classifiers with deep features, Experimental found that AlexNet excels as a feature extractor, while SVM demonstrates superiority as a classifier because of this combination achieving the highest accuracy rates of 99.0% and 95.16% for the CK+ database and the JAFFE datasets, respectively.

# INTRODUCTION

In recent years, data gathering has reached a peak with cameras installed in the public network, and data are being gathered and kept up for better outcomes for particular tasks. Facial expressions, such as happiness, disgust, sadness, fear, surprise, and anger are nonverbal communication modalities that form a plentiful portion of the collecting data. Automatic facial expression recognition (FER) is a difficult challenge; however, it can

Corresponding author
Doaa Alebiary,
doaa.mohamed@fci.bu.edu.eg

achieve practical solutions for various applications in real life. Accurately understanding a human's emotion is a challenging topic that facial computer vision and related approaches are trying to automate and solve (*Huang et al., 2019*).

In general, FER is accomplished through four key modules: image improvement, face detection, feature extraction, and classification. Statistical models or template-based matching are used to detect faces, and the image improvement is carried out using image processing methods like filtering, wavelet transforms, noise reduction, *etc*., (*Saurav, Saini & Singh, 2021*). The feature extraction technique extracts look, shape, texture, motion, landmarks, geometry, and other local and global properties of facial parts. Finally, classification using either a supervised or unsupervised method is conducted. To classify or cluster the characteristics, there are a variety of accessible classifiers and clustering algorithms (*Akhand et al., 2021*). In addressing expression recognition as a classification challenge, conventional approaches frequently rely on manually crafted features like local binary patterns (LBP) and traditional machine learning techniques such as support vector machine (SVM) for classification. While these methods may demonstrate effectiveness with datasets gathered under controlled laboratory conditions, they struggle to perform well with more complex expression datasets collected in uncontrolled environments. Fortunately, the advent of deep learning has marked a significant breakthrough in terms of both ease of use and effectiveness, particularly in addressing image classification tasks (*Li et al., 2020*).

Several studies have found that deep learning or convolutional neural networks (CNNs) exceed handmade techniques. However, training a deep network from the start is costly in terms of computing and time, and it may necessitate a huge dataset. To overcome this problem, we can use a pre-trained network. The lower layers of a pre-trained network are already trained for recognizing different shapes and sizes. Then, the upper layers can be refined for dealing with the new target data set, and this technique is known as transfer learning (*Gupta, Arunachalam & Balakrishnan, 2020*). There are many pre-trained networks available now, including AlexNet (*Sekaran, Lee & Lim, 2021*), GoogleNet (*Shaees et al., 2020*), ResNet (*He et al., 2016*), and DeepNet (*Nunes et al., 2016*). These networks can be leveraged to achieve different special goals of the different datasets.

## Motivation

Facial expression recognition holds significant promise and relevance in various domains, ranging from technology and psychology to sociology and artificial intelligence. By enabling machines to understand and interpret human emotions, facial expression recognition facilitates more intuitive and empathetic human-machine interactions and enhances our understanding of human behavior and cognition.

## Contributions

The main contributions of this article include proposing two different methods to solve FER as the following:

1) Use pre-trained deep convolutional neural networks (AlexNet, GoogleNet, ResNet-18, ResNet-50, and ResNet-101) as feature extractors and use these features as input to

traditional classifiers (SVM, ensemble bagging, K-nearest neighbor (KNN), and naïve Bayes) to recognize the facial expression.

2) Fine-tune the pre-trained deep convolutional neural networks (DCNNs) (AlexNet, GoogleNet, and ResNet-18) by using transfer learning, which is achieved by altering the last fully connected layer with a new one influenced by the dataset's class number.

The rest of this article is structured as follows, the literature review is detailed in the following section, while "The Proposed Method" presents the proposed method including the employed techniques. The datasets, results, experiments, and comparisons with recent methods are presented in "Experimental Results". The conclusion of the article is presented in "Conclusion".

## LITERATURE REVIEW

The research goal in the area of facial expression recognition was to identify human feelings based on image or video records. The traditional approaches extracted features from a face image and then classify emotion based on the value of those aspects. Deep learning-based approaches address the FER problem by combining both processes into a unified approach. Numerous publications have scrutinized and contrasted existing FER methodologies, with recent studies predominantly focusing on deep learning-based techniques. Below, we provide a succinct overview of the methodologies employed in the most prominent FER approaches.

### FER methodologies based on machine learning

In the realm of artificial intelligence (AI), especially within the subdomain of machine learning, automatic FER poses a formidable challenge. Across the evolution of the FER problem, various classic machine-learning approaches have been employed.

*Xiao-xu & Wei (2007)* presented a facial expression recognition (FER) technique utilizing the wavelet energy feature (WEF) to enhance recognition accuracy. Initially, WEF is applied to the facial image, followed by feature extraction using Fisher's linear discriminants (FLD). Subsequently, emotions are classified using the KNN approach. Additionally, principal component analysis (PCA) and non-negative matrix factorization (NMF) are employed, and the authors in *Zhao, Zhuang & Xu (2008)* utilized KNN for classification in FER. *Feng, Pietikäinen & Hadid (2007)* employed a method where they extracted local binary pattern histograms from various small regions of the image. These histograms were then combined into a feature histogram. Subsequently, they utilized a linear programming (LP) approach to evaluate emotions by classifying the histograms based on their corresponding emotional categories.

*Shih, Chuang & Wang (2008)* examined various features, including principal component analysis (PCA) and discrete wavelet transform (DWT). They concluded that the integration of DWT with 2D-linear discriminant analysis (LDA) yielded superior performance compared to other methods. *Shan, Gong & McOwan (2009)* conducted a comprehensive investigation using different SVM variations to explore alternative facial representations based on local binary patterns (LBPs) and local statistical characteristics.

*Jabid, Kabir & Chae (2010)* explored an appearance-based technique known as a local directional pattern (LDP). In *Alshamsi, Kepuska & Meng (2017)*, SVM was employed to examine two feature descriptors: the center of gravity descriptor and the facial landmarks descriptor. In *Liew & Yairi (2015)*, SVM, along with numerous other methods such as linear discriminant analysis (LDA) and KNN, was evaluated for classifying features extracted through various methods, including Gabor filters, Haar wavelets, and LBP.

## FER methodologies based on deep learning

*Zhao, Shi & Zhang (2015)* presented a novel approach by integrating a deep belief network (DBN) with a neural network (NN) for FER. The DBN was utilized for unsupervised feature learning, whereas the NN was employed for emotion feature classification.

*Pranav et al. (2020)* employed a typical CNN construction with featuring two convolutional-pooling layers. Conversely, *Mollahosseini, Chan & Mahoor (2016)* investigated a more intricate design incorporating two convolutional-pooling layers and four inception layers. Different CNNs were trained using varied filter sizes in the convolutional layers, as outlined in *Pons & Masip (2017)*.

In *Wen et al. (2017)*, the authors employed a group of CNNs, training 100 CNNs, and only a subset of them were utilized in the final model. Similarly, *Ruiz-Garcia et al. (2017)* utilized CNNs trained with facial images and initialized the weights with encoder weights of layered convolutional auto-encoder, which proved to outperform CNNs with random initialization. Furthermore, In *Ding, Zhou & Chellappa (2017)*, the authors introduced the FaceNet2ExpNet architecture, which extends deep facial recognition architecture to facial expression recognition (FER). In *Chirra, Uyyala & Kolli (2021)*, using a multi-block DCNN setup, we introduced two models utilizing ensemble learning methods. The first model combines a bagging ensemble with SVM (DCNN-SVM), while the second model utilizes an ensemble of three distinct classifiers with a voting technique (DCNN-VC). The architecture presented by *Gera & Balasubramanian (2021)* employs a novel spatial-channel attention net (SCAN) to obtain local and global attention for each channel at every spatial location. By converting the input data into RGB and depth map images, and subsequently utilizing a recursive procedure with randomized channel concatenation, *Behzad et al. (2021)* demonstrated a sparsity-aware deep network capable of automatic recognition of 3D/4D facial expressions. *Hernández-Luquin & Escalante (2021)* introduced an improved CNN-based structure that integrates multiple branches consisting of radial basis function (RBF) units. In *Kar et al. (2021)*, the authors introduced a novel method for accurately recognizing facial expressions using a combination of a hybrid feature descriptor and an enhanced classifier. Drawing inspiration from the effectiveness of the stationary wavelet transform in various computer vision applications, the technique initially applies the stationary wavelet transform to preprocess the facial image. Subsequently, the pyramid of histograms of orientation gradient features is computed from the low-frequency stationary wavelet transform coefficients. This step aims to extract more prominent details from facial images, thereby enhancing the recognition process. *Kim et al. (2022)* proposed a novel approach to facial expression recognition, employing a hybrid model that merges CNNs with a SVM classifier, utilizing dynamic facial expression data.

The method involves utilizing dense facial motion flows and geometry landmark flows extracted from facial expression sequences as inputs for the CNN and SVM classifier, respectively. Additionally, CNN architectures tailored for facial expression recognition based on dense facial motion flows are proposed. *Shaik & Cherukuri (2022)* delved into the topic of recognizing facial expressions using a deep learning technique known as the Visual-Attention-based Composite Dense Neural Network (VA-CDNN).

## THE PROPOSED METHOD

The FER problem has been addressed using a variety of strategies, however, this problem remains a great challenge, so new methods are still needed to solve it. Algorithm 1 illustrates an algorithm for the proposed method (FERDCNN). The main three components of the proposed method are pre-processing data, feature extraction, and the classification of seven emotions: neutral, sadness, happiness, disgust, fear, anger, and surprise. The framework for the proposed method (FERDCNN) is presented in Fig. 1.

The first portion involved pre-processing techniques like face detection then cropping, resizing, gamma correction, and histogram equalization method. The second part uses pre-trained CNN models for feature extraction. The third part executes the classification task, and two approaches are used the first one is based on a traditional classifier for classification. The second approach uses fine-tuning of a pre-trained deep CNN model and is performed by using transfer learning to classify according to the number of classes present in the datasets under examination. The proposed method is fully explained in the next sub-sections.

### The pre-processing

There are four steps in the pre-processing part: face detection and cropping, resize, gamma correction, and histogram equalization as shown in Fig. 2.

The face detection step is carried on by using the Haar algorithm, It aims to remove background and non-facial areas, and then crop the facial area. The next step in the pre-processing is resizing the resolution to the pre-trained model's default input size, *i.e.*, we used AlexNet which has an input size equal to $[227, 227]$ while GoogleNet, ResNet-18, ResNet-50, and ResNet-101 have an input size equal to $[224, 224]$.

The third step is Gamma correction, also known as contrast adjustment, which is a technique within the photographic toolkit that is utilized to modify the displayed image, which is defined in Eq. (1):

$$I_{out} = AI_{in}^{\gamma} \tag{1}$$

where the non-negative real input value $I_{out}$ is raised to the power $\gamma$ and multiplied by the constant A = 1 to get the output value $I_{in}$. Any value between 0 and infinity can be assigned to Gamma. When Gamma is set to 1 (the default value), the mapping is linear. For values less than 1, Gamma is weighted towards higher (brighter) output values, while for values greater than 1, the mapping is weighted towards lower (darker) output values, We tested our system with three distinct Gamma correction options, and we found that the highest accuracy was achieved at a Gamma value of 1.7 shown in Table 1.

**Algorithm 1 FERDCNN method.**

**Input:** Image data and corresponding image label from the expression dataset.

**Output:** The recognition accuracy.

1: Start

2: Pre-process images according to the dimension required for the input layer of DCNN, crop face, gamma correction, and then equalize the image.

3: Extract deep features for all images in the dataset

4: Using Classifiers to classify each image.

5: Similarly, extract features from the test set also using DCNN.

6: Use the newly trained classifier to predict the labels for the test set.

7: Calculate the recognition rate by comparing the output of DCNN with the image label.

8: Display the recognition accuracy.

9: End Start

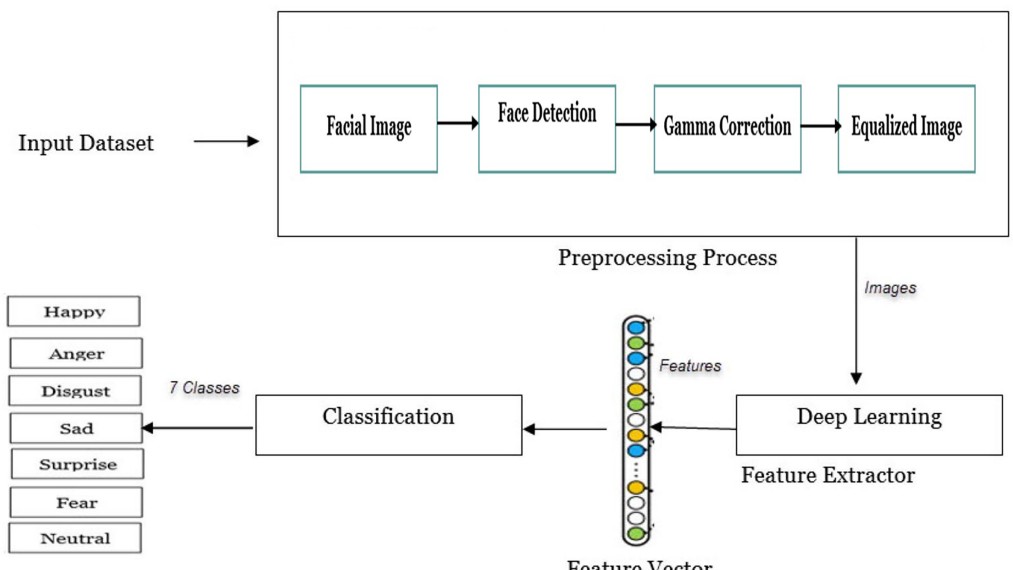

**Figure 1 The framework for the proposed method (FERDCNN) of facial expression recognition.**

Normalizing the image's grey scale value and improving the ability to distinguish between the brightness of the foreground and background in a facial image can both be done with histogram equalization (Hist-eq), so it is used as the fourth step in the pre-processing part. The histogram function is shown in Eq. (2):

$$H(v) = \frac{cdf(v) - cdf_{min}}{(M \times N) - cdf_{min}} \times (L - 1) \tag{2}$$

where $H(v)$ represents the histogram function of the resized $n$-number of face images, $cdf(v)$ denotes the cumulative distribution function, $cdf_{min}$ specifies the minimum non-

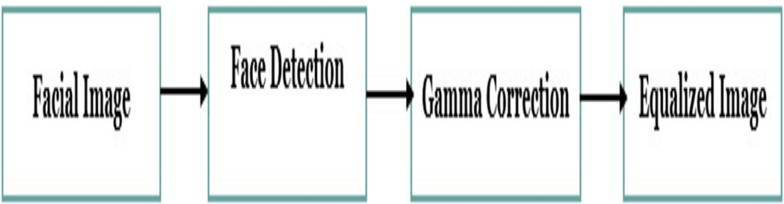

**Figure 2** The steps for the pre-processing phase.     

**Table 1** Testing three distinct gamma correction options.

| Dataset | Gamma = 1.7 | Gamma = 1 | Gamma = 0.5 |
| --- | --- | --- | --- |
| CK+ | 99.0 | 96.3 | 95.2 |
| JAFFE | 95.15 | 91.5 | 89.9 |
| KDEF | 73.72 | 70.5 | 69.2 |

zero value of the cumulative distribution function, $M \times N$ indicates the total number of pixels in the image where $M$ represents the width, $N$ denotes the height, and $L$ defines the number of greyscale image levels.

## The deep features extraction

FER is one of the computer vision applications that largely use the DCNN. Convolutional layers, pooling layers, and fully connected layers are the three different types of layers that make up a DCNN. Training and testing DCNN require enough computational power and huge training samples. To optimize this, deep feature extraction can be implemented through pre-trained deep models. Five pre-trained CNN models (AlexNet, GoogleNet, ResNet-18, ResNet-50, ResNet101) are used and tested in the proposed method.

## The classification approaches

As mentioned before two different classification approaches are tested in the proposed method. The first approach uses traditional supervised classifiers, and the second approach uses deep classifiers. The deep and traditional classifiers and their parameter setup are briefly covered in this section.

### The traditional supervised classifier approaches

Numerous traditional supervised machine learning (ML) classifiers exist, that can be trained using DCNN this work considers four classifiers: SVM, ensemble bagging, KNN, and naive Bayes.

- **Support vector machine (SVM)** (*Hsu & Lin, 2002*) was initially designed for binary classification but can be adapted for multi-class classification tasks. Support vectors are a portion of the training set used to determine where the separation hyperplane is located. Using SVM, the predictor will grow more exact in proportion to the complication of the data.

- **Ensemble (bagging)** (*Bühlmann, 2012*) ensemble creates ensemble decision trees. Tree Bagger selects an arbitrary sample of indicators to apply in each decision split equivalent to a random forest. With the use of bootstrap-aggregated decision trees, the impacts of over-fitting are minimized and generalization is enhanced. In the proposed method, 100 trees are used while training the tree bagger.
- **K-nearest neighbors (KNN)** (*Dino & Abdulrazzaq, 2019*) is a pattern recognition algorithm that uses training datasets to find the k nearest relatives in future examples. The developer chooses the k neighbours in an experimental manner.
- **Naive Bayes** (*Leung, 2007*) performs the classification by measuring the possibility of whether a data point belongs inside a specific category or not.

### *The deep classifier approaches*

In this classifier, the proposed method trains three deep classifiers using transfer learning (AlexNet, GoogleNet, and ResNet-18). The final fully connected layer neurons of the pre-trained DCNN are adjusted to align with the specific number of classes required for the current classification task.

To fine-tune DCNN for FER, three different training optimizers are used, Adaptive Moment Estimation (adam) (*Kingma & Ba, 2015*), stochastic gradient descent with momentum (sgdm) (*Liu, Gao & Yin, 2020*), and root mean square propagation (rmsprop) (*Dogo et al., 2018*). The learning rate serves as a critical hyper-parameter, governing the pace at which a neural network is trained by determining the speed at which weights are adjusted in response to predicted errors. Identifying the optimal learning rate is often challenging and requires significant time and effort. Overly high rates can result in rapid but unsteady training, while low rates typically require lengthy training periods and may even get stuck before being completed successfully. Each deep model is trained with a batch size of 128 and a learning rate variation of 0.01, $t$0.001, 0.0001, 0.00001.

## EXPERIMENTAL RESULTS

The proposed method, including pre-processing and the DCNN, is implemented using MATLAB on a DELL PC with the following specifications: Processor Intel(R) Core(TM) i7-9750H, CPU @ 2.60 GHz, 2.59 GHz, RAM: 16 GB, and Windows 11 (64-bit) as the operating system. The proposed method is evaluated for performance using the CK+ and JAFFE datasets, as detailed in "Datasets", while "Recognition Performance with Feature Extractor and Traditional Classifier Approaches" and "The Effectiveness of the Proposed Method Utilizing DCNN as a Feature Extractor and Deep Classifier Approaches" discuss the experimental results and a comparison with other recent methods.

### Datasets

The presented method was trained and tested using the CK+ (*Kanade, Cohn & Tian, 2000*), JAFFE (*Lyons et al., 1998*) and KDEF (*Lundqvist, Flykt & Ohman, 1998*) datasets. The fundamental emotion dataset has seven classes: anger, disgust, fear, happiness, sadness, and surprise besides neutrality for CK+ and contempt for JAFFE. The datasets

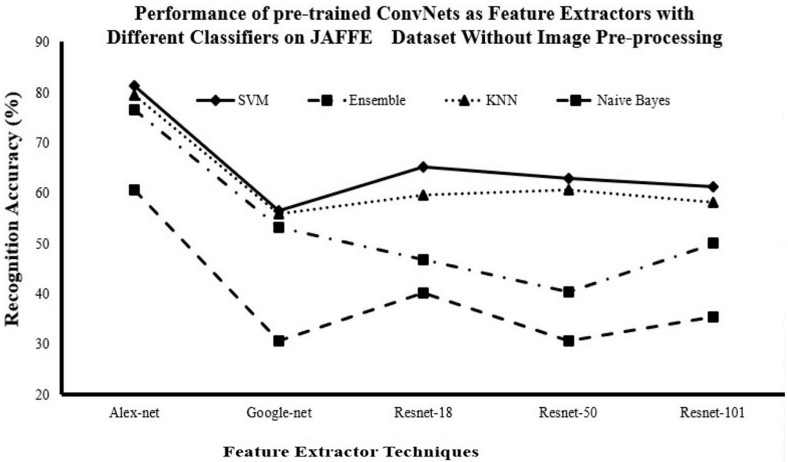

**Figure 3 Comparison of recognition accuracy for DCNN as features extractor and traditional supervised classifiers without image pre-processing on JAFFE dataset.**

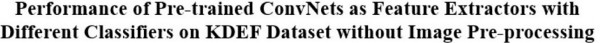
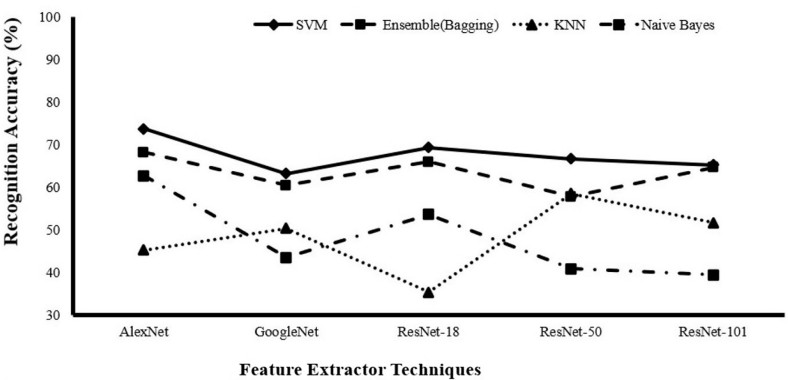

**Figure 4 Comparison of recognition accuracy for DCNN as features extractor and traditional supervised classifiers without image pre-processing on the KDEF dataset.**

were randomly partitioned into training and testing sets, with 80% of the data allocated for training and 20% for testing. This division aims to enhance the performance of the proposed method for facial expression recognition.

## Recognition performance with feature extractor and traditional classifier approaches

Four different traditional supervised classifiers are tested with deep features, like the features retrieved by five DCNN (AlexNet, GoogleNet, ResNet-18, ResNet-50, and ResNet-101) with and without image pre-processing to evaluate the performance of the proposed method, and these trials are applied on JAFFE and CK+ datasets. The recognition accuracy with and without pre-processing is fully explained in the next sub-sections.

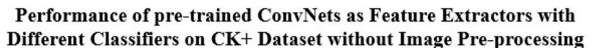

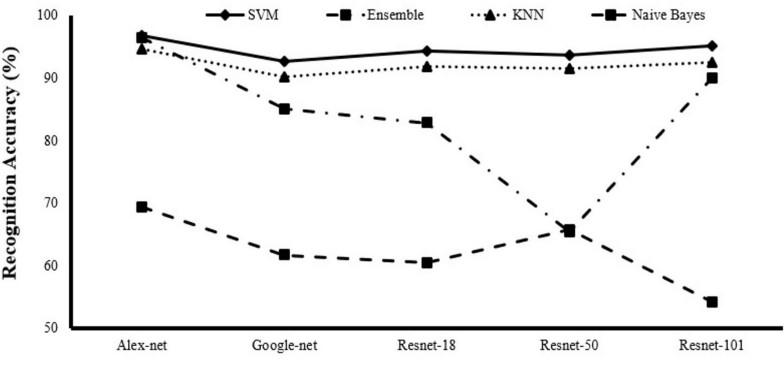

**Figure 5 Comparison of recognition accuracy for DCNN as features extractor and traditional supervised classifiers without image pre-processing on the CK++ dataset.**

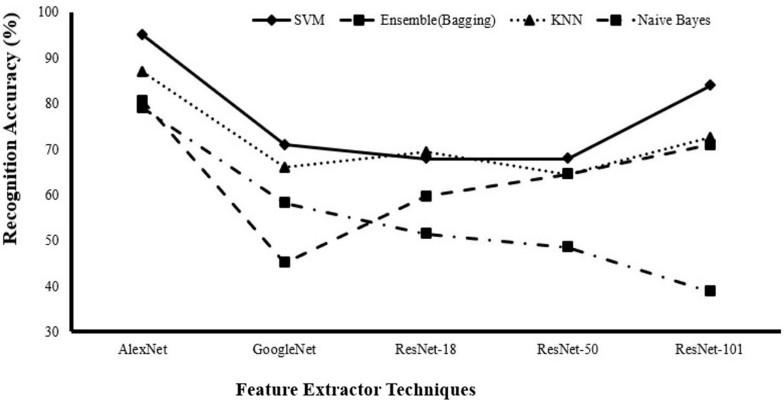

**Figure 6 Comparison of recognition accuracy for DCNN as features extractor and traditional supervised classifiers after image pre-processing on the JAFFE dataset.**

### *Recognition accuracy for DCNN as features extractor and traditional supervised classifiers without image pre-processing*

Figures 3–5 show that AlexNet with all classifiers achieves the highest performance. And SVM with all networks achieves the highest performance. So, it is clear that AlexNet excels as a feature extractor, while SVM demonstrates superiority as a classifier because of this combination achieving the highest accuracy with 81.29%, 73.72% and 95.7% for JAFFE, KDEF and CK+ datasets respectively.

### *Recognition accuracy for DCNN as features extractor and traditional supervised classifiers with image pre-processing*

Figures 6–8 show the enhancement in the recognition accuracy after using image pre-processing on JAFFE, KDEF and CK+ datasets. They show that AlexNet with all classifiers

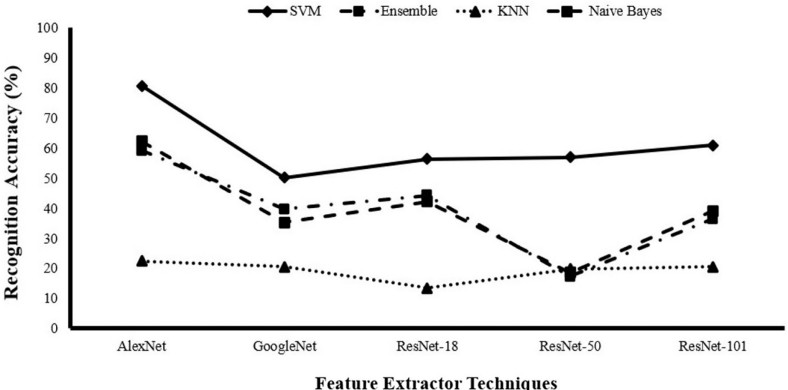

**Figure 7 Comparison of recognition accuracy for DCNN as features extractor and traditional supervised classifiers after image pre-processing on the KDEF dataset.**

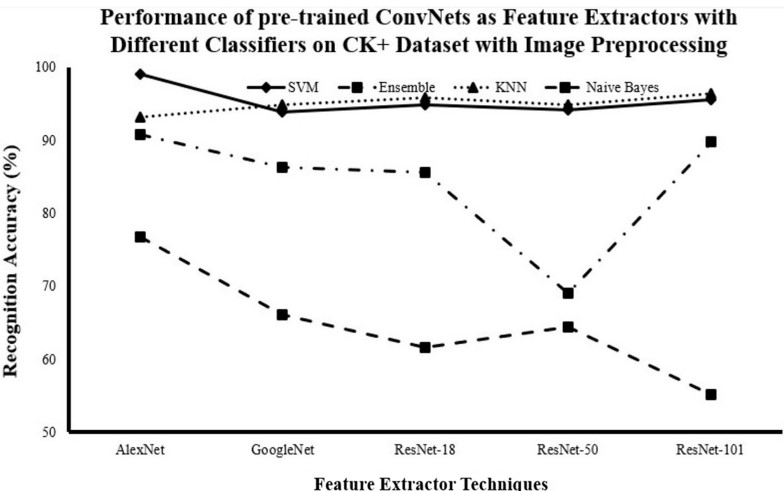

**Figure 8 Comparison of recognition accuracy for DCNN as features extractor and traditional supervised classifiers after image pre-processing on the CK++ dataset.**

achieves the highest performance and that SVM with all networks achieves the highest performance. Therefore, the combination of AlexNet excels as a feature extractor, while SVM demonstrates superiority as a classifier because of this combination by achieving the highest accuracy with 95.16%, 81.31% and 99.00% for the JAFFE, KDEF and CK+ datasets, respectively.

### The confusion matrices

The confusion matrix keeps track of the values in the true classification and predicted classification columns for each item. To create balance, rows are used to display the actual classifications while columns are used to provide model predictions. The confusion

**Table 2 The confusion matrix of classification on the JAFFE dataset.**

| TRUE | Predicted | | | | | | | Accuracy(%) | Recall(%) | Precision(%) |
|---|---|---|---|---|---|---|---|---|---|---|
| | Sad | Anger | Disgust | Fear | Happiness | Neutral | Surprise | | | |
| Sad | 9 | 0 | 0 | 0 | 0 | 0 | 0 | 100 | 100 | 90 |
| Anger | 0 | 9 | 0 | 0 | 0 | 0 | 0 | 100 | 100 | 100 |
| Disgust | 1 | 0 | 6 | 1 | 0 | 0 | 0 | 96.43 | 75 | 100 |
| Fear | 0 | 0 | 0 | 10 | 0 | 0 | 0 | 100 | 100 | 90.9 |
| Happiness | 0 | 0 | 0 | 0 | 8 | 0 | 0 | 100 | 100 | 100 |
| Neutral | 0 | 0 | 0 | 0 | 0 | 9 | 0 | 100 | 100 | 90 |
| Surprise | 0 | 0 | 0 | 0 | 0 | 1 | 8 | 98.41 | 88.9 | 100 |

**Table 3 The confusion matrix of classification on the CK+ dataset.**

| TRUE | Predicted | | | | | | | Accuracy (%) | Recall (%) | Precision (%) |
|---|---|---|---|---|---|---|---|---|---|---|
| | Sad | Anger | Disgust | Fear | Happiness | Contempt | Surprise | | | |
| Sad | 40 | 0 | 0 | 0 | 0 | 3 | 0 | 93 | 100 | 93.7 |
| Anger | 0 | 16 | 0 | 0 | 0 | 0 | 0 | 100 | 100 | 100 |
| Disgust | 3 | 0 | 53 | 0 | 0 | 0 | 0 | 100 | 100 | 100 |
| Fear | 0 | 0 | 0 | 22 | 0 | 0 | 0 | 100 | 100 | 100 |
| Happiness | 0 | 0 | 0 | 0 | 62 | 0 | 0 | 100 | 100 | 100 |
| Contempt | 0 | 0 | 0 | 0 | 0 | 22 | 0 | 100 | 88 | 100 |
| Surprise | 0 | 0 | 0 | 0 | 0 | 0 | 74 | 100 | 100 | 100 |

matrices of the proposed method (FERDCNN) on the JAFFE and CK+ datasets are presented in Tables 2 and 3 respectively.

In fact, it is evident from the confusion matrices which classes were incorrectly assigned to different classes if these classes share a trait that would explain the errors, or whether the fault was noticeable during network training.

We conducted some statistical analysis by running the system ten times, calculating the maximum, minimum, average, and standard deviation values shown in Table 4. We noticed that the minimum value in the first case is greater than any value in the other cases, and similarly, the standard deviation in the first case is the lowest value, indicating the stability and strength of this system.

## The effectiveness of the proposed method utilizing DCNN as a feature extractor and deep classifier approaches

Three separate training optimizers sgdm, adam, and rmsprop, are used to train neural networks. In order to fine-tune the deep models using the JAFFE and CK+ datasets, several learning rates and training functions were used in the experiments detailed in Tables 5 and 6. The AlexNet with Adam and a learning rate of 0.0001 achieves the greatest

**Table 4 Statistical analysis of proposed method on the JAFFE dataset**

| Model | Lowest run | Best run | Average accuracy | Std |
|---|---|---|---|---|
| AlexNet+SVM | 94.2 | 96.3 | 95.1 | 0.621825 |
| AlexNet+Ensemble | 66.5 | 83.9 | 76.18 | 6.353792 |
| AlexNet+Naive Bayes | 38.7 | 66.1 | 50.84 | 8.507017 |
| AlexNet+kNN | 77.4 | 91.9 | 85.8 | 4.608205 |

**Table 5 Effectivness of deep classifier by changing the optimizer and learning rate on the JAFFE dataset.**

| Classifier | Optimizer | Learning rate | | | |
|---|---|---|---|---|---|
| | | 0.01 | 0.001 | 0.0001 | 0.00001 |
| | | Recognition accuracy | | | |
| AlexNet | adam | 20 | 35 | **95** | 85 |
| | sgmd | 15 | 80 | 70 | 50 |
| | rmsprop | 15 | 20 | 90 | 94 |
| GoogleNet | adam | 14.63 | 14.63 | 80.49 | 63.41 |
| | sgmd | 14.63 | 92.68 | 71.43 | 33.33 |
| | rmsprop | 14.29 | 14.63 | 85.37 | 76.19 |
| ResNet-18 | adam | 30.29 | 92.86 | 93.86 | 83.33 |
| | sgmd | 85.71 | 88.1 | 69.52 | 21.43 |
| | rmsprop | 19.05 | 90.48 | 90.48 | 76.19 |

Note:
The best result is shown in bold.

**Table 6 Effectiveness of deep classifier by changing the optimizer and learning rate on the CK+ dataset.**

| Classifier | Optimizer | Learning rate | | | |
|---|---|---|---|---|---|
| | | 0.01 | 0.001 | 0.0001 | 0.00001 |
| | | Recognition accuracy | | | |
| AlexNet | adam | 21 | 30.5 | 97 | 86.15 |
| | sgmd | 20.24 | 90 | 80 | 60 |
| | rmsprop | 30.24 | 50.52 | 97.94 | **100** |
| AlexNet | adam | 25.13 | 94.87 | 97.94 | 96.92 |
| | sgmd | 25.13 | 95.88 | 95.38 | 96.41 |
| | rmsprop | 18.56 | 82.52 | 97.94 | 100 |
| ResNet-18 | adam | 30.29 | 92.86 | 93.86 | 83.33 |
| | sgmd | 85.71 | 88.1 | 69.52 | 21.43 |
| | rmsprop | 19.05 | 90.48 | 90.48 | 76.19 |

Note:
The best result is shown in bold.

recognition accuracy of 95% for JAFFE dataset, and the AlexNet with rmsprop and a learning rate of 0.00001 achieves the greatest recognition accuracy of 100% for CK+ dataset.

**Table 7 Compare the recognition performance of the suggested method with recent methods on the JAFFE and CK+ datasets.**

| Method | Year | Method name | JAFFE | CK+ |
|---|---|---|---|---|
| *Gera & Balasubramanian (2021)* | 2021 | Spatio-channel attention net (SCAN) | 58.49 | 91.4 |
| *Behzad et al. (2021)* | 2021 | Sparsity-aware deep learning | – | 97.28 |
| *Fu et al. (2020)* | 2020 | Neighborhood semantic transformation | – | 98.58 |
| *Hernández-Luquin & Escalante (2021)* | 2021 | Multi-branch deep radial basis function | 95.83 | 98.58 |
| *Li et al. (2020)* | 2020 | Attention mechanism-based CNN | 98.52 | 98.68 |
| *Shima & Omori (2018)* | 2018 | CNN+SVM | 95.31 | – |
| *Minaee, Minaei & Abdolrashidi (2021)* | 2021 | Attentional convolutional network | 92.8 | 98 |
| *Niu, Gao & Guo (2021)* | 2021 | Oriented FAST and rotated BRIEF (ORB) and LBP | 90.5 | 96.2 |
| *Shaik & Cherukuri (2022)* | 2022 | Visual attention based composite dense neural network (VA-CDNN) | 97.67 | 97.46 |
| *Gowda & Suresh (2022)* | 2022 | Active Learning and SVM | 88.31 | – |
| *Borgalli & Surve (2022)* | 2022 | Custom CNN | 95 | 83 |
| | | DCNN (AlexNet) + Traditional Classifier (SVM) | 95.16 | 99 |
| Proposed method (FERDCNN) | 2022 | DCNN (AlexNet) with adam optimizer | 95 | 95 |
| | | DCNN (AlexNet) with rmsprop optimizer | 94 | 100 |

**Table 8 Compare the recognition performance of the suggested method with recent methods on the KDEF datasets.**

| Method | Year | Method name | KDEF |
|---|---|---|---|
| *Yaddaden, Adda & Bouzouane (2021)* | 2021 | LBP and HOG | 85.48 |
| *Kas et al. (2021)* | 2021 | Combining textural and shape | 76.73 |
| *Kas et al. (2021)* | 2019 | Hand-crafted and learned feature extraction | 77.86 |
| Proposed method | 2024 | DCNN (AlexNet) + Traditional Classifier (SVM) | 81.3 |

## Comparison of the recognition performance of the proposed method with the state-of-the-art

Table 7 illustrated the overall performance of the proposed method compared to other deep learning recent methods on the CK+ and JAFFE datasets, and Table 8 illustrated the overall performance of the proposed method compared to other deep learning recent methods on the KDEF dataset. The proposed method has achieved a competitive performance in FER in comparison with previous methods. The proposed method achieves first place in the CK+ dataset and fifth place in the JAFFE dataset with accuracy very similar to the third and fourth places with a difference in the accuracy of no more than 0.67% with the third place.

## DISCUSSION

FER has caught the research community's attention recently because it can affect many real-life applications. FER is accomplished through four key modules: image improvement, face detection, feature extraction, and classification. This article proposes an efficient

method for FER (FERDCNN) verified on five different pre-trained CDNN models. The presented method was trained and tested using the CK+ and JAFFE datasets, the fundamental emotion dataset has seven classes: anger, disgust, fear, happiness, sadness, and surprise besides neutrality for CK+ and contempt for JAFFE. The proposed method provides two approaches for FER.

The first approach uses conventional classifiers and deep features extracted by pre-trained networks, four different traditional supervised classifiers (SVM, ensemble (bagging), KNN and naïve Bayes) are tested with deep features, like the features retrieved by five DCNN (AlexNet, GoogleNet, ResNet-18, ResNet-50, and ResNet-101) with and without image pre-processing to measure the performance of the proposed method. We noticed that AlexNet excels as a feature extractor, an SVM demonstrates superiority as a classifier.

In the second approach, the proposed method was applied with three pre-trained DCNNs (AlexNet, GoogleNet, ResNet-18) trained using transfer learning. Three separate optimizers sgdm, adam, and rmsprop, are used to train neural networks. As shown in Tables 5 and 6. The AlexNet with Adam and a learning rate of 0.0001 achieves the greatest recognition accuracy for JAFFE dataset, and the AlexNet with rmsprop and a learning rate of 0.00001 achieves the greatest recognition accuracy for the CK+ dataset.

## CONCLUSION

This article performed experiments to classify facial emotion expression into seven classes of emotions. The experiments were run on two datasets JAFFE and CK+. The proposed method provides two approaches for FER. The first approach uses traditional supervised classifiers trained on deep features extracted by AlexNet, GoogleNet, ResNet-18, ResNet-50, and ResNet-101 pre-trained networks. This approach achieved the highest accuracy with the AlexNet feature extractor and the SVM classifier which equal 99.00% and 95.16% accuracy on the CK+, and JAFFE datasets, respectively. In the second approach, the proposed method was applied with three pre-trained deep CNNs trained using transfer learning. This approach achieves recognition accuracy with 100% and 95% on the CK+, and JAFFE datasets, respectively.

The experimental results conclude that transfer learning through fine-tuning is more accurate than the other previous works that used the same dataset. The proposed method may be improved by trying to create custom two-dimensional convolutional neural networks (2D-CNN) and increasing the dataset to improve the recognition efficiency.

Future research will enhance the method by incorporating additional diverse features, such as speech and motion, to further bolster its robustness. Our current method is exclusively applicable to 2D images. Moving forward, we aim to refine our architecture to accommodate video data, 3D face datasets, as well as depth images, and explore more effective machine-learning techniques to augment the network's performance.

### Funding

This work was supported by the Deanship of Scientific Research, Prince Sattam bin Abdulaziz University, Al-Kharj, Saudi Arabia *via* funding from Prince sattam bin Abdulaziz University project number (PSAU/2024/R/1445). The funders had no role in study design, data collection and analysis, decision to publish, or preparation of the manuscript.

### Grant Disclosures

The following grant information was disclosed by the authors:
Deanship of Scientific Research, Prince Sattam bin Abdulaziz University.
Al-Kharj, Saudi Arabia *via* funding from Prince sattam bin Abdulaziz University project number: PSAU/2024/R/1445.

### Competing Interests

The authors declare that they have no competing interests.

### Author Contributions

- Metwally Rashad conceived and designed the experiments, performed the experiments, performed the computation work, authored or reviewed drafts of the article, and approved the final draft.
- Doaa Alebiary conceived and designed the experiments, performed the experiments, analyzed the data, performed the computation work, prepared figures and/or tables, authored or reviewed drafts of the article, and approved the final draft.
- Mohammed Aldawsari analyzed the data, prepared figures and/or tables, authored or reviewed drafts of the article, and approved the final draft.
- Ahmed Elsawy performed the experiments, prepared figures and/or tables, authored or reviewed drafts of the article, and approved the final draft.
- Ahmed H. AbuEl-Atta conceived and designed the experiments, performed the experiments, performed the computation work, prepared figures and/or tables, and approved the final draft.

### Data Availability

The Facial Expression Recognition software is available at Zenodo: Alebiary, D. (2024). FERDCNN_Facial Expression Recognition. Zenodo. https://doi.org/10.5281/zenodo.11204565.

The data set (CK+, KDEF, and JAFFE) is available at Kaggle and Zenodo:

- https://www.kaggle.com/datasets/davilsena/ckdataset

- Lyons, M., Kamachi, M., & Gyoba, J. (1998). The Japanese Female Facial Expression (JAFFE) Dataset [Data set]. Zenodo. https://doi.org/10.5281/zenodo.3451524.

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
