# Peer review of "FERDCNN: an efficient method for facial expression recognition through deep convolutional neural networks"

_PeerJ Computer Science, doi:10.7717/peerj-cs.2272_

## Round 0.1 · original submission · Major Revisions

The reviewers have given several critical comments on the paper for improvement. The authors should significantly improve the quality of the paper by carefully addressing the reviewer's comments.

**Language Note:** The review process has identified that the English language must be improved. PeerJ can provide language editing services - please contact us at [email protected] for pricing (be sure to provide your manuscript number and title). Alternatively, you should make your own arrangements to improve the language quality and provide details in your response letter. – PeerJ Staff

Reviewer 1 ·

Basic reporting

To further improve the basic reporting, the authors should consider addressing the following areas:

Introduction Detail: The introduction should be expanded to provide more justification for the study, specifically by elaborating on the knowledge gap being filled. This will help to enhance the context and significance of the research.

Language Improvement: The English language should be improved to ensure clarity and comprehension for an international audience. Specific examples where language could be improved have been highlighted, and it is suggested that a proficient English speaker with subject matter expertise review the manuscript or professional editing services be utilized.

Statistical Analysis: While the manuscript is commended for its professional and unambiguous language, the weakness in the statistical analysis should be addressed and improved before acceptance.

Experimental design

no comment

Validity of the findings

To improve the validity of the findings, the authors should consider assessing the impact and novelty of their research, clearly articulating the rationale and benefits to the literature. Additionally, providing a more detailed description of the rationale behind the study and its potential contributions to the field would enhance the validity of the findings.

Additional comments

How does the proposed FERDCNN method compare to other state-of-the-art facial expression recognition techniques in terms of computational efficiency, scalability, and robustness in handling diverse datasets and real-world scenarios?

What are the potential limitations or challenges in implementing the proposed method in practical applications, and how could these be addressed to enhance the method's applicability and effectiveness?

Have the ethical considerations and implications of the research been thoroughly addressed, particularly in terms of privacy and data security when dealing with facial images? Are there any specific measures taken to ensure ethical and responsible use of the data?

What are the potential future research directions or extensions of the proposed method, and how could they contribute to advancing the field of facial expression recognition, particularly in addressing emerging challenges or incorporating multi-modal data sources?

How does the proposed method address potential biases or inaccuracies in facial expression recognition, particularly in diverse and multicultural settings? Are there specific strategies or considerations for ensuring fairness and accuracy in recognizing expressions across different demographic groups?

Are there any specific recommendations for the practical implementation of the proposed method in real-world scenarios, such as in human-computer interaction systems or emotion recognition technologies? How could the method be adapted or optimized for specific application domains or user groups?

Reviewer 2 ·

Basic reporting

.

Experimental design

All are correct

Validity of the findings

Reviewer Comments:
Overall, I could say that I enjoyed reading this work entitled " An Efficient Method for Facial Expression Recognition Through Deep Convolutional Neural Networks". I would suggest some comments though, which are the following:

(1) The author should try again to improve English expressions in the text. Although the use of English is 100% clear, the author should try to improve some parts to make the text closer to a native speaking audience.

(2) In this paper, I feel that the statistical analysis is very weak, so I suggest that the authors conduct Wilcoxon statistical tests, Friedman statistical tests, and ANOVA tests.
The authors should compare their experimental results with some recent techniques published in 2023-2024.

(3) The motivation and contribution of this study are unclear. So, authors should provide motivation and contribution in a separate section.

(4) As part of the conclusion section, authors should include the drawbacks of the proposed method and also be sure to explain how the drawbacks will be overcome in the future.

(5) Abstracts should mention some experimental results as well.

Reviewer 3 ·

Basic reporting

The paper is well organized, but there are some points to include in this paper.
1. Discuss clearly with a diagram about face preprocessing steps that will be your Figure 3.
2. The effects of several gamma corrections must be shown on the facial digital images.
3. Eq. 1 is not correct. Make it well-defined with more justifications.
4. The employed CNN architectures mention no clear parameter listing and no. of blocks.
5. The aspect ratio of Fig, 6, 7, 8 must be maintained.
6. Enrich the literature review with some recent facial expression papers.

Experimental design

1. Your facial expression recognition system's accuracy must be shown by the results corresponding to images corrected by Three Distinct Gamma Correction Options.
2. Use the KDEF facial expression dataset also and show the results.

Validity of the findings

1. There should be a conclusion section, whereas the discussion should be merged into the results section and given a new subsection.
1. Why Normalizing the image’s grey scale value? Show this with some image examples.

---

## Round 0.2 · accepted · Accept

I am pleased to inform you that your paper has been evaluated positively by the reviewers and is now ready for acceptance. The manuscript demonstrates a high level of scholarly rigor and contributes significantly to the field.

Before we proceed with the final acceptance, we kindly request you to address the remaining minor comments provided by the reviewers. Ensuring these last details are taken care of will further enhance the clarity and impact of your work.

Reviewer 2 ·

Basic reporting

All review responses have been completed properly. In my opinion, this paper has been accepted for publication.

Experimental design

Very good

Validity of the findings

Good

Reviewer 3 ·

Basic reporting

1. The changes done in Man_Track.pdf are good.
2. There should be correct alignment and sequencing of paragraphs with good quality of images in the accepted version of the manuscript.

Experimental design

The experiments are now statifying the work.

Validity of the findings

The outcomes are experimentally validated.